# CADGrasp: Learning Contact and Collision Aware General Dexterous Grasping in Cluttered Scenes

**Jiyao Zhang**[1,2][*] **, Zhiyuan Ma**[1,2][*] **, Tianhao Wu**[1,2]**, Zeyuan Chen**[1,2]**, Hao Dong**[1,2][†]

[1] Center on Frontiers of Computing Studies, School of Computer Science, Peking University
[2] National Key Laboratory for Multimedia Information Processing,
School of Computer Science, Peking University
jiyaozhang@stu.pku.edu.cn

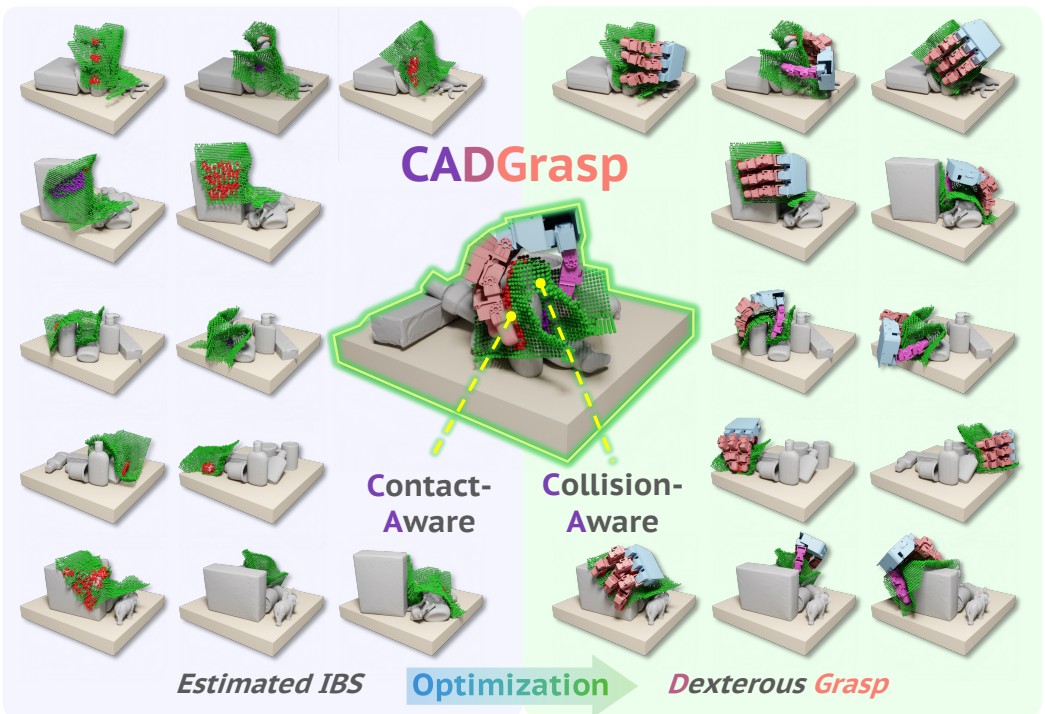

Figure 1: We propose **CADGrasp**, which learns a contact- and collision-aware intermediate representation as a constraint, and further obtains the dexterous grasp pose with an optimization method to achieve single-view dexterous hand grasping in cluttered scenes.

## Abstract

Dexterous grasping in cluttered environments presents substantial challenges due to the high degrees of freedom of dexterous hands, occlusion, and potential collisions arising from diverse object geometries and complex layouts. To address these challenges, we propose **CADGrasp**, a two-stage algorithm for general dexterous grasping using single-view point cloud inputs. In the first stage, we predict

*: equal contribution, †: corresponding author

39th Conference on Neural Information Processing Systems (NeurIPS 2025).

sparse IBS, a scene-decoupled, contact- and collision-aware representation, as the optimization target. Sparse IBS compactly encodes the geometric and contact relationships between the dexterous hand and the scene, enabling stable and collision-free dexterous grasp pose optimization. To enhance the prediction of this high-dimensional representation, we introduce an occupancy-diffusion model with voxel-level conditional guidance and force closure score filtering. In the second stage, we develop several energy functions and ranking strategies for optimization based on sparse IBS to generate high-quality dexterous grasp poses. Extensive experiments in both simulated and real-world settings validate the effectiveness of our approach, demonstrating its capability to mitigate collisions while maintaining a high grasp success rate across diverse objects and complex scenes. More details and videos are available at `https://ibsgrasp.github.io/`.

## 1 Introduction

Dexterous grasping in cluttered scenes is a critical step toward enabling a dexterous hand to autonomously perform diverse tasks in real-world environments. Compared to single-object dexterous grasping [1, 2, 3, 4, 5, 6, 7], the diverse and complex layouts in cluttered scenes [8, 9, 10, 11, 12] introduce additional challenges beyond generalizing to various objects. The stacking of objects leads to occlusion, resulting in partial observations without full object geometry. Moreover, the restricted graspability caused by stacking requires more precise grasp poses while avoiding collisions with surrounding objects to prevent unintended outcomes, such as target object displacement due to contact between the dexterous hand and nearby objects.

Current methods focus on constructing large-scale synthetic datasets to capture the distribution of potential cluttered scenes [10, 11]. Based on these datasets, existing approaches typically first filter scene points with high graspability. Conditioned on these points, they employ either regression-based [10] or generative-based models [11] to predict grasp poses. However, directly mapping partial point cloud observations to grasp poses is challenging to generalize due to the non-linearity of the mapping from 3D point cloud space to pose space and the sensitivity of physical constraints to small variations in hand pose [13]. Current methods [14, 15, 16, 13] adopt a two-stage framework that combines contact map prediction and optimization to enhance generalization. However, these methods are primarily designed for single-object grasping, assume access to complete object geometry for optimization, which is not available in real-world cluttered scenes due to partial observation.

To tackle this problem, we propose predicting a contact- and collision-aware intermediate representation to serve as the optimization target as shown in Figure 1. The proposed representation, sparse IBS, is the interaction bisector surface (IBS) [17] between the scene and the dexterous hand, incorporating compact contact indicators. This representation is decoupled from the scene, eliminating the requirement for complete scene geometry, making it well-suited for cluttered scenes. Additionally, sparse IBS captures both geometric and contact information between the scene and the dexterous hand, making it effective for optimizing stable and collision-free dexterous grasp poses. To efficiently generate such a high-dimensional representation, we propose an occupancy-diffusion model with voxel-level conditional guidance and force closure score filtering. To further obtain stable and collision-free dexterous grasp poses, we design a set of energy functions tailored to sparse IBS.

In our experiments, we conduct extensive evaluations in simulation environments featuring 670 diverse cluttered scenes containing over 1300 objects. Comparative results demonstrate the effectiveness of our framework, while ablation studies further validate the effectiveness of our design. Our analysis highlights the stability and collision-awareness of the generated grasp poses. Additionally, we evaluate our method against other baselines in real-world settings to validate its practicality.

In summary, our contribution is as follows:

- We propose a two-stage framework consisting of scene-decoupled, contact- and collision-aware intermediate representation prediction and constrained grasp pose optimization for general dexterous grasping in cluttered scenes.

- We propose an occupancy-diffusion model with voxel-level conditional guidance and force closure score filtering to enhance representation prediction, along with several energy functions and ranking strategies to improve final grasp pose optimization.

- We conduct comprehensive simulation and real-world experiments to demonstrate the effectiveness of our method.

## 2 Related Work

### 2.1 One-stage Dexterous Grasp Pose Prediction

One-stage dexterous grasp pose generation [1, 2, 3, 18, 19, 20] aims to train an end-to-end model to predict grasp poses. Regression-based methods [19, 20, 21] assume a one-to-one mapping between the object or scene and the grasp pose, which is limited in capturing the multi-modal dexterous grasp pose distribution due to the high degree of freedom. Generative-based methods [18, 2, 22, 23, 24] can model complex distributions, making them more suitable for dexterous grasp pose generation. Current approaches leverage physical constraints [3, 2], such as contact and penetration, to enhance grasp pose quality. However, end-to-end methods struggle with generalization due to the non-linearity between the observation space and pose space, as well as the sensitivity of physical constraints to small errors in grasp poses [13]. This challenge becomes more pronounced in cluttered environments, where the diversity of objects and layouts necessitates extremely large-scale grasp pose datasets [10, 11], limiting generalization. In contrast, we propose a two-stage framework that first predicts an intermediate representation in the observation space, followed by optimization based on this representation to enhance generalization.

### 2.2 Two-stage Dexterous Grasp Pose Prediction

Two-stage dexterous grasp pose generation [16, 14, 25, 26, 27, 28] decomposes grasp pose prediction into two stages to mitigate the challenges of direct mapping. Typically, the first stage predicts either the grasp pose [25, 26, 27, 29, 30] or an intermediate representation [28, 16, 14], followed by optimization based on physical constraints between the object and the dexterous hand. However, these methods are specifically designed for single-object grasping and assume prior knowledge of complete object geometries, which is often unavailable in cluttered scenes due to object stacking [11, 10], even with multi-view cameras. This limitation restricts the applicability of two-stage methods in cluttered environments. In contrast, we propose a sparse IBS representation that serves as an intermediate representation, enabling the application of a two-stage framework in cluttered scenes.

### 2.3 Hand-Object Representation

Hand-object representation can be primarily utilized in two ways. The first is to serve as the observation [31, 32, 28, 33], compressing the observation feature space and reducing the complexity of geometric feature learning, thereby enhancing grasp pose generation [28] or grasp policy learning [31, 32]. The second is to act as an intermediate representation for two-stage dexterous grasp pose generation methods [14, 16]. The most common intermediate representation for two-stage grasp pose generation is the contact map [16, 15, 13], which computes the distance between each point on the object and the dexterous hand. However, this representation requires complete object geometry, which is not accessible in cluttered scenes. The most relevant to our work is Interaction Bisector Surface (IBS) [32], which computes a surface between the scene and the dexterous hand along with various spatial and contact information. However, such representation has only been used as an observation representation [32, 33]. In contrast, we propose using IBS as an intermediate representation for grasp pose optimization. To reduce the difficulty of predicting such representation, we design a more compact sparse IBS representation and develop a specialized module to predict sparse IBS along with optimization strategies tailored to our sparse IBS representation.

## 3 Method

**Problem Formulation.** In this work, we consider the problem of generating a set of grasp poses for a dexterous hand to grasp objects in a cluttered environment. We define a grasp pose of a dexterous hand as a tuple $\mathbf{g} = \{\mathbf{T}, \mathbf{J}\}$, where $\mathbf{T} \in \mathrm{SE}(3)$ indicates the wrist pose, $\mathbf{J} \in \mathbb{R}^n$ is the joint configuration of the hand, and $n$ is the degree of freedom (DoF) of the hand. Given a single view pointcloud $\mathcal{P} \in \mathbb{R}^{N \times 3}$ of a cluttered scene, we estimate the grasp poses $\mathcal{G} = \{\mathbf{g}_i\}_{i=1}^{|\mathcal{G}|}$ that are stable and collision-free for dexterous grasping.

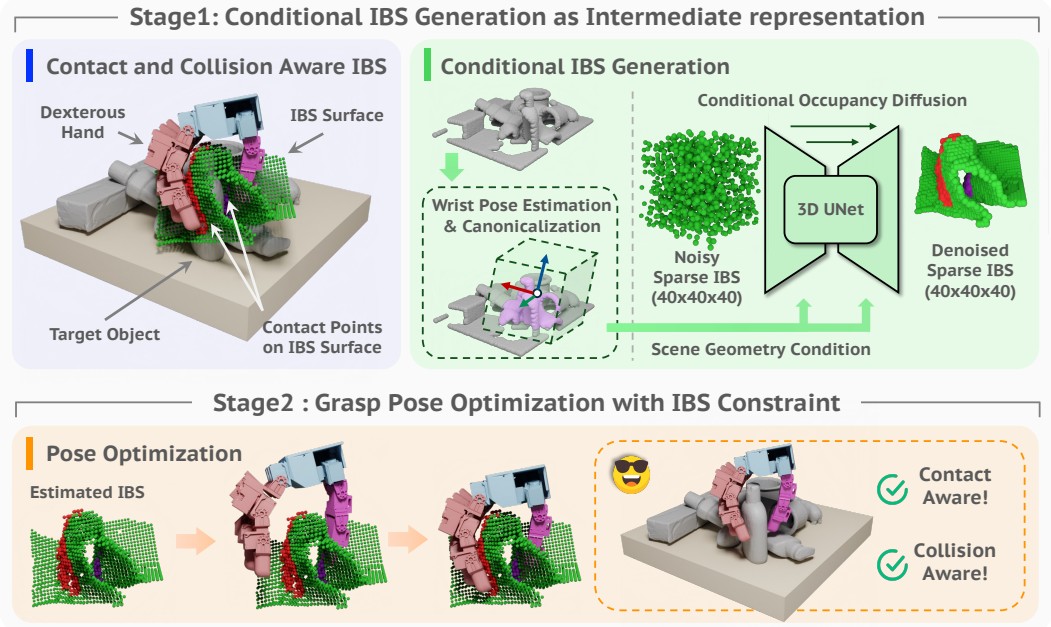

Figure 2: **Overview of CADGrasp**, a two-stage framework for dexterous grasping in cluttered scenes. **(I)** Conditional IBS Generation: A diffusion model is trained to model the conditional probability distribution $p(\mathcal{I}|\mathcal{P}, \mathbf{T})$. **(II)** Grasp Pose Optimization: We optimize the grasp poses $\mathcal{G}$ with predicted sparse IBS $\hat{\mathcal{I}}$ as constraints.

**Overview.** The overview of our method is shown in Figure 2. We propose a two-stage framework for dexterous grasping in cluttered scenes, called **CADGrasp**. We use the sparse IBS $\mathcal{I}$ that is aware of contact and collisions as an intermediate representation between the two stages, which can efficiently encode the geometric relationship between the dexterous hand and the scene corresponding to the successful grasp poses ( Section 3.1). In the first stage, we model the conditional probability distribution $p(\mathcal{I}|\mathcal{P}, \mathbf{T})$ by a diffusion model, where $\mathcal{I}$ is the sparse IBS, $\mathcal{P}$ is the single-view observed scene point cloud, and $\mathbf{T}$ is the wrist pose ( Section 3.2). In the second stage, with the predicted sparse IBS $\hat{\mathcal{I}}$ as constraints, we get dexterous grasp poses $\mathcal{G}$ via an optimization algorithm ( Section 3.3).

## 3.1 Contact and Collision Aware IBS for Dexterous Grasping.

IBS [17] is the Voronoi diagram between two close 3D geometric objects. Inspired by the efficiency of IBS in describing the spatial relationship between 3D objects and its successful application in dexterous hand manipulation [32, 33], we adapt IBS to represent the geometric relationship between the dexterous hand and the environment as well as the object when a successful grasping state is achieved. We generate the sparse IBS $\mathcal{I}$ in a simulator for training, as shown in Figure 3. Specifically, for each dexterous grasp pose $\mathbf{g}$, we define a canonical space centered at the grasp seed point $p_s$ with the rotation $\mathbf{R}$ of the wrist pose $\mathbf{T} = [\mathbf{R}|\mathbf{t}]$ as the direction. The sparse IBS $\mathcal{I} \in \mathbb{R}^{n \times n \times n \times 3}$ is defined with a resolution of $n$, where the last three dimensions are:

- Occupancy of the IBS surface, which is 1 if the voxel in $\mathcal{I}$ in IBS surface, otherwise it is $-1$.

- Occupancy of contact points between the thumb and the target object, which is 1 if the voxel in $\mathcal{I}$ is a contact point between the thumb and the object, otherwise it is $-1$.

- Occupancy of contact points between other fingers and the object, which is 1 if the voxel in $\mathcal{I}$ is a contact point between other fingers and the object, otherwise it is $-1$.

The sparse IBS $\mathcal{I}$ encodes the geometric relationship between the hand corresponding to a successful grasp pose $\mathbf{g}$ and the environment, which includes not only the configuration of the hand but also the

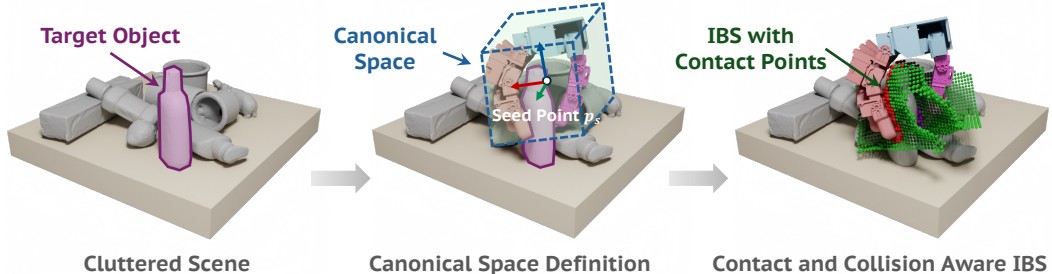

**Figure 3: Creation of the sparse IBS for dexterous grasping.** Given a cluttered scene, we first generate the grasp pose **g** using an optimization algorithm. Then, we canonicalize and crop the scene point cloud $\mathcal{P}$ to obtain the canonicalized point cloud $\mathcal{P}^*$. Finally, we compute the sparse IBS $\mathcal{I}$ based on the canonicalized point cloud.

contact relationship between fingers and objects, and specifies a safety bound to ensure collision-free with environments.

### 3.2   Conditional IBS Generation.

In this section, our goal is to generate the intermediate representation IBS to provide constraints for the subsequent grasp pose optimization stage. Specifically, as shown in Figure 2, given a single-view observed point cloud $\mathcal{P}$, we first predict the wrist pose **T** of the dexterous hand. Then, given **T** and $\mathcal{P}$, we generate multiple IBS candidates $\{\mathcal{I}_i\}_{i=1}^n$, where $n$ is the number of candidates. Finally, we rank the IBS candidates and select the optimal one $\hat{\mathcal{I}}$ as the final intermediate representation.

**Wrist Pose Estimation.**   We use the same structure as DexGraspNet2.0 [11]. First, given the scene point cloud $\mathcal{P} \in \mathbb{R}^{N \times 3}$, where $N$ is the number of points, we extract point-wise features $\mathcal{F} = \{f_i\}_{i=1}^N$ using ResUNet14 [34] and predict point-wise graspness $\mathcal{S} = \{s_i\}_{i=1}^N$. Finally, after ranking and FPS sampling, we obtain the final set of grasp seed points $\{p_s^i\}_{i=1}^M$, where $M$ is the number of sampled points. For each grasp seed point $p_s$, we condition on the corresponding point feature $f$, and use a denoising diffusion model [35] to directly model the joint probability distribution $p(\mathbf{T}|f)$ in Euclidean space and obtain the denoised wrist pose **T** with reverse ODE process.

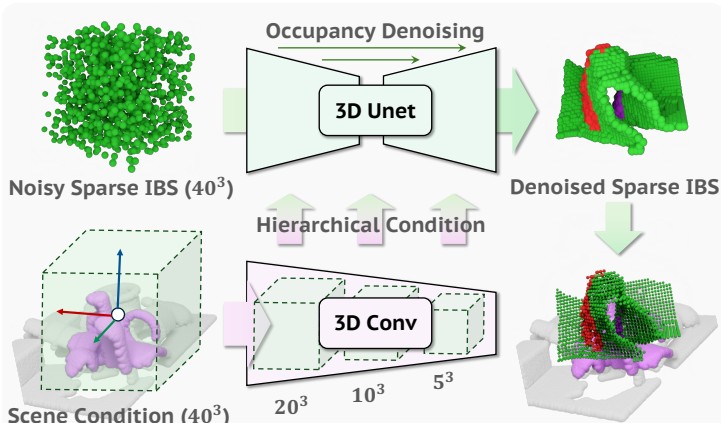

**Figure 4: IBS generation.** We train a conditional occupancy-diffusion model to model the conditional probability distribution $p(\mathcal{I}|\mathcal{P}^*)$, where $\mathcal{P}^*$ is the canonicalized and voxelized point cloud. The voxel-level alignment provides hierarchical conditions during the denoising process.

**IBS Candidates Generation.** As shown in Figure 4, given the selected grasp seed point $p_s$ and the corresponding wrist pose **T**, we define a canonical space with $p_s$ as the coordinate origin and the rotation of **T** as the direction. We canonicalize and voxelize the original observed point cloud $\mathcal{P}$

to obtain $\mathcal{P}^* \in \mathbb{R}^{n \times n \times n \times 1}$, where $n$ is the voxelization resolution. The above operations simplify the modeling of the probability distribution $p(\mathcal{I}|\mathcal{P}, \mathbf{T})$ to modeling $p(\mathcal{I}|\mathcal{P}^*)$, making the feature space more compact and reducing the training difficulty of the network. Specifically, inspired by the dominance of denoising diffusion models in 3D generation [36, 35], we model $p(\mathcal{I}|\mathcal{P}^*)$ based on an occupancy-diffusion model [35]. The $\mathcal{I} \in \mathbb{R}^{n \times n \times n \times 3}$ has the same resolution as $\mathcal{P}^*$, and this voxel-level alignment provides hierarchical conditions during the generation of IBS, improving both the generation quality and efficiency. As shown in Figure 4, for both occupancy network $\Omega_o$ and point cloud network $\Omega_p$, we use 3D UNet as the backbone network. UNet has 4 levels: $40^3$, $20^3$, $10^3$, $5^3$, with feature dimensions of 32, 64, 128, and 256 respectively at each level. We achieve voxel-level condition guidance by concatenating the point cloud features at corresponding levels to the occupancy features. The training of the diffusion network uses the following loss:

$$\mathcal{L}_{\mathcal{I}_0} = \mathbb{E}_{\epsilon \sim \mathcal{N}(0,I), t \sim \mathcal{U}(0,1)} \left\| \Omega_o(\mathcal{I}_t, t, \Omega_p(\mathcal{P}^*)) - \mathcal{I}_0 \right\|_2^2 \tag{1}$$

where $\epsilon$ and $\mathcal{I}_0$ are the data sample and $\mathcal{I}_t$ is the noisy sample at time step $t$. $\mathcal{N}$ is the Gaussian distribution, and $\mathcal{U}$ is the uniform distribution.

**IBS Ranking.** Considering that the sampling process has a certain probability of sampling in low-density regions, we sample multiple IBS candidates $\{\hat{\mathcal{I}}_i\}_{i=1}^m$ from the estimated distribution $p(\mathcal{I}|\mathcal{P}^*)$, where $m$ is the number of samples. Furthermore, we calculate the force closure score $\{\mathcal{Q}_{\hat{\mathcal{I}}_i}\}_{i=1}^m$ for candidates based on the contact points and directions of the thumb and other fingers, and obtain a ranked sequence of IBS candidates $\hat{\mathcal{I}}_{\sigma_0} \succ \hat{\mathcal{I}}_{\sigma_1} \succ \cdots \succ \hat{\mathcal{I}}_{\sigma_m}$, where:

$$\hat{\mathcal{I}}_{\sigma_i} \succ \hat{\mathcal{I}}_{\sigma_j} \iff \mathcal{Q}_{\hat{\mathcal{I}}_{\sigma_i}} > \mathcal{Q}_{\hat{\mathcal{I}}_{\sigma_j}} \tag{2}$$

Finally, we select the top-ranked IBS as the final intermediate representation $\hat{\mathcal{I}}$.

### 3.3 Grasp Pose Optimization with IBS Constraints.

**Grasp Pose Optimization.** Given the predicted IBS $\hat{\mathcal{I}}$, we generate the dexterous grasp poses $\mathcal{G}$ through an optimization algorithm. Specifically, we use a gradient-based optimization algorithm to minimize the energy function $\mathbf{E}$, which consists of four parts: 1) joint limits energy $\mathbf{E}_j$, 2) self-penetration energy $\mathbf{E}_{sp}$, 3) contact energy $\mathbf{E}_d$, and 4) collision energy $\mathbf{E}_p$. We obtain the dexterous grasp poses $\mathcal{G}$ by minimizing the energy function. Specifically:

$\mathbf{E}_j$ is used to limit the joint angles within a preset range, defined as:

$$\mathbf{E}_j = \frac{1}{d} \sum_{i=1}^{d} \left( \max(\theta_i - \theta_i^{\max}, 0) + \max(\theta_i^{\min} - \theta_i, 0) \right) \tag{3}$$

where $\theta_i$ is the angle of the $i$-th joint, $\theta_i^{\max}$ and $\theta_i^{\min}$ are the maximum and minimum angles of the $i$-th joint, respectively, and $d$ is the number of joints.

$\mathbf{E}_{sp}$ is used to limit the self-penetration of the hand, defined as:

$$\mathbf{E}_{sp} = \frac{1}{|\mathcal{P}_h|^2} \sum_{p \in \mathcal{P}_h} \sum_{q \in \mathcal{P}_h} [p \neq q] \max(\delta - d(p, q), 0) \tag{4}$$

where $\mathcal{P}_h$ is the set of points on the hand, $d(\cdot)$ calculates the Euclidean distance between two points, and $\delta$ is the safety distance for self-penetration of the hand.

$\mathbf{E}_p$ is used to constrain the contact relationship between fingers and objects, defined as:

$$\mathbf{E}_p = \frac{1}{|\mathcal{P}_h|} \sum_{p \in \mathcal{P}_h} \max \left( 0, -\left( \frac{p - p^*}{\|p - p^*\|} \cdot \mathbf{n} \right) \right) \tag{5}$$

where $\mathcal{P}_h$ is the set of points on the hand, $p$ is the point on the hand, $p^*$ is the nearest point in the IBS point set $\mathcal{P}_{\mathcal{I}}$ corresponding to $p$, and $\mathbf{n}$ is the normal vector of the estimated IBS surface at point $p^*$.

$\mathbf{E}_d$ is used to constrain the optimization of the hand in the safety space without collision with the environment, defined as:

$$\mathbf{E}_d = \frac{\alpha_1}{|\mathcal{P}_t^*|} \sum_{p^* \in \mathcal{P}_t^*} \min_{p \in \mathcal{P}_t} \|p^* - p\|^2 + \frac{\alpha_2}{|\mathcal{P}_o^*|} \sum_{p^* \in \mathcal{P}_o^*} \min_{p \in \mathcal{P}_o} \|p^* - p\|^2 + \frac{\alpha_3}{|\mathcal{P}_t| + |\mathcal{P}_o|} \sum_{p \in \mathcal{P}_t, \mathcal{P}_o} \min_{p^* \in \mathcal{P}_t^*, \mathcal{P}_o^*} \|p^* - p\|^2 \tag{6}$$

where $\mathcal{P}_t^*$ and $\mathcal{P}_o^*$ are the sets of points on the IBS that are in contact with the thumb and other fingers, respectively, $\mathcal{P}_t$ and $\mathcal{P}_o$ are the sets of points on the thumb and other fingers, respectively. $\alpha_1$, $\alpha_2$, and $\alpha_3$ are hyperparameters used to balance the weights between different energy terms.

Overall, the final energy function is:

$$\mathbf{E} = \lambda_1 \mathbf{E}_j + \lambda_2 \mathbf{E}_{\text{sp}} + \lambda_3 \mathbf{E}_p + \lambda_4 \mathbf{E}_d \tag{7}$$

where $\lambda_1$, $\lambda_2$, $\lambda_3$, and $\lambda_4$ are the weights of each energy term.

**Grasp Pose Ranking.** Given the inherent uncertainties in the optimization process, we simultaneously optimize multiple grasp configurations $\mathcal{G}$ based on a predicted sparse IBS $\hat{\mathcal{I}}$. We record the optimization residuals $\{\mathbf{E}_{\mathbf{g}_i}\}_{i=1}^{k}$, where $k$ denotes the number of optimization trials. The grasp configurations $\mathcal{G}$ are then ranked according to their residuals, and the configuration with the minimal residual is selected as the optimal grasp pose $\mathbf{g}$.

## 4 Experiments

### 4.1 Experimental Setup

**Datasets.** We use the same object datasets and simulation environments as DexGraspNet 2.0 [11]. The object datasets consist of 60 training objects from GraspNet1Billion [8] and 1259 testing objects from GraspNet1Billion and ShapeNet [37]. We sample 100 scenes from the 7600 training scenes of DexGraspNet2 [11] for the training of the IBS generation module. And we use the full 670 scenes from the testing set of DexGraspNet2 [11] for the testing. The testing scenes are categorized into three density levels: loose, random, and dense.

**Baselines.** We compare our method with the following dexterous grasp pose prediction methods.

- **DexGraspNet2.0[11]:** An end-to-end diffusion-based pipeline for grasp pose generation in cluttered scenes, directly mapping 3D point cloud observations to grasp poses.
- **HGC-Net[10]:** A regression-based model for direct pose prediction in cluttered scenes.
- **ISAGrasp[38]:** This baseline is originally designed for single-object grasping using an end-to-end regression-based model. Following the adaptation strategy in DexGraspNet2.0 [11], we extend ISAGrasp to cluttered scenes.
- **GraspTTA[14]:** A two-stage, single-object grasping framework that needs complete point clouds for the second-stage optimization, which is impractical in cluttered scenes; we therefore adapt it for cluttered scenes as DexGraspNet2.0 does and remove the optimization stage. This two-stage framework is originally designed for single-object grasping and requires complete point clouds for second-stage optimization, which is not feasible in cluttered scenes. Therefore, following DexGraspNet2.0 [11], we adapt GraspTTA for cluttered scenes and omit the optimization stage.

**Evaluation in Simulation.** We report the **Success Rate**, following the same evaluation protocol as DexGraspNet2.0 [11]. In the simulation, we evaluate from two perspectives:

- **Object and Scene Generalization.** We evaluate the performance of our method following the same protocol as DexGraspNet2.0 [11].
- **Cross-Embodiment Generalization.** Different from other baselines, our method can directly zero-shot generalize to unseen embodiments, thanks to the proposed universal intermediate representation. We evaluate our method using an unseen dexterous hand (Allegro) in a zero-shot manner.

**Real-world Setup.** As shown in Figure 5, we conduct real-world dexterous grasping experiments in clutter scenes, using a Flexiv Rizon-4 robot arm equipped with a Leap Hand [39] as the end-effector.

A third-person view RealSense D415 camera is employed for perception. We selected 30 objects with diverse shapes, sizes, and materials, as depicted in Figure 6. Following the simulation experiment setup, we also assess grasping under varying levels of clutter density, covering the entire object dataset across 5 cluttered scenes with 4 to 8 objects per scene, as illustrated in Figure 6. For each scene, the policy continues grasping until two consecutive failures occur. For each grasp execution, we follow the same sequence as in the simulation: pre-grasp, grasp, squeeze fingers, and lift. We report the **Success Rate** as the number of successfully grasped objects divided by the total number of attempts.

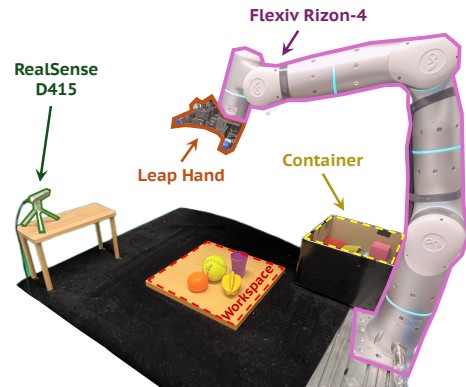

Figure 5: **Real-world experiment setup.**

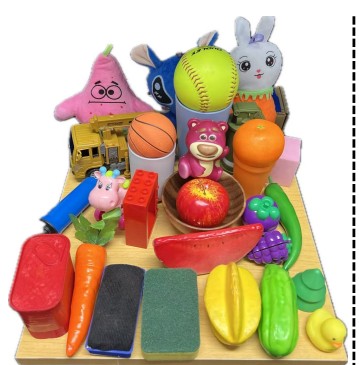 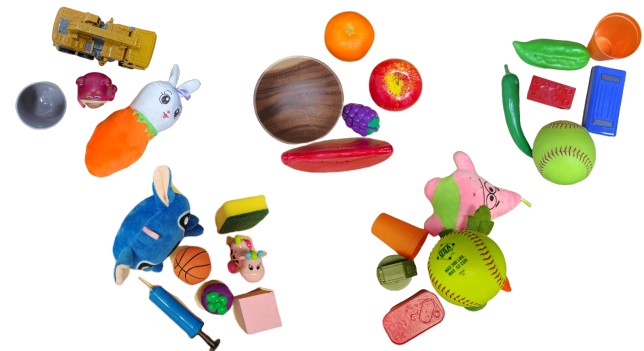

Figure 6: **Real-world object datasets and evaluated cluttered scenes.** The left image shows the objects used in real-world testing. The right image shows the layouts of objects in five different test scenes, with 4 to 8 objects per scene.

| Method | Ratio | GraspNet-1Billion | | | ShapeNet | | |
|---|---|---|---|---|---|---|---|
| | | Dense | Random | Loose | Dense | Random | Loose |
| **HGC-Net**[†] [10] | 1 | 46.0 | 37.8 | 26.7 | 46.4 | 44.8 | 30.4 |
| **GraspTTA**[†] [14] | 1 | 62.5 | 54.1 | 42.8 | 56.6 | 57.8 | 46.4 |
| **ISAGrasp**[†] [38] | 1 | 63.4 | 60.7 | 51.4 | 64.0 | 56.3 | 52.7 |
| **DexGraspNet2.0** [11] | 1/1000 | 83.3 | 79.5 | 73.9 | **81.5** | 77.1 | 73.5 |
| **Ours** | 1/1000 | **86.5** | **85.5** | **80.1** | 79.3 | **77.7** | **75.7** |
| **Ours (Allegro)** | 1/1000 | 77.6 | 75.0 | 74.9 | 75.7 | 76.6 | 73.0 |

Table 1: **Comparison results.** [†] indicates the results are from [11]. **Ratio** refers to the ratio of the number of grasps for training compared to the whole dataset. Each **Dense** scene contains 8-11 objects, and each **Random** scene contains 1-10 objects, obtained by deleting objects from Dense scenes, and each **Loose** scene contains 1-2 objects. **Allegro** is the result evaluated with the Allegro, and others use the Leap hand for evaluation. We report only our method, trained on Leap Hand and tested on Allegro Hand, since other methods do not demonstrate cross-hand generalization.

## 4.2 Simulation Results

**Comparison with Baselines.** As shown in Table 1, the regression-based methods HGC-Net and ISAGrasp struggle with the complex dexterous grasp pose distribution. The generative-based method GraspTTA performs even worse, likely due to the absence of a second-stage optimization step. DexGraspNet2.0 achieves a higher success rate by leveraging the diffusion model. However, its direct

end-to-end mapping is susceptible to challenges associated with non-linear mapping and sensitivity to physical constraints. In contrast, our two-stage approach demonstrates superior performance. Furthermore, to evaluate robustness, we report success rates with standard deviations over 20 random seeds. The consistently small deviations indicate stable performance across initializations.

**Cross-Embodiment Generalization.** Since our proposed sparse IBS representation is embodiment-agnostic, we directly use the same generated sparse IBS that was used to evaluate the Leap Hand to optimize the Allegro Hand grasp pose. As shown in Table 1, the Allegro Hand also achieves comparable results to the Leap Hand, demonstrating the universality of our proposed representation.

## 4.3 Real-world Results

As shown in Table2, our method achieves an average grasp success rate of 93.3%, significantly surpassing the baseline at 83.9%. While the two methods exhibit comparable performance on medium-scale objects, our approach demonstrates markedly greater robustness for small and flat objects, in line with the simulation analyses reported in Table7. This improvement can be attributed to the effectiveness of the IBS. By integrating IBS into the optimization, our method explicitly avoids collisions with the table and surrounding objects, yielding safer grasps, which is particularly critical in densely cluttered scenes. The observed real-world performance underscores the practical applicability of our approach for deployment in real robotic systems.

| Method | Scene #1 | Scene #2 | Scene #3 | Scene #4 | Scene #5 | Overall |
|---|---|---|---|---|---|---|
| DexGraspNet2.0 [11] | **100.0% (4/4)** | 50.0% (2/4) | 71.4% 5/7) | 100.0% (7/7) | **88.9% (8/9)** | 83.9% (26/31) |
| **CADGrasp (Ours)** | **100.0% (4/4)** | **100.0% (5/5)** | **85.7% (6/7)** | **100% (7/7)** | **88.9% (8/9)** | **93.8% (30/32)** |

Table 2: **Real-world results.** We evaluated 30 objects across 5 real-world cluttered scenes. In each scene, the policy continues attempting to grasp until two consecutive failures occur. The number to the left of '/' indicates the number of successful grasps, while the number to the right indicates the total number of grasp attempts.

## 4.4 Computational Efficiency

We assess the inference efficiency of our method on a single NVIDIA RTX 4090 GPU by averaging 50 independent runs to suppress stochastic fluctuations in the sampling-based optimization. A detailed breakdown of the runtime is reported in Table 3. The end-to-end time to generate a single grasp is 6.51 s on average, measured without any task-specific engineering optimizations. While not real-time, our method delivers a substantial latency advantage over existing state-of-the-art two-stage

| Module | Stage | Time (s) |
|---|---|---|
| Wrist Pose Estimation | 1 | 1.38 |
| IBS Generation | 1 | 1.39 |
| IBS Ranking | 1 | 0.71 |
| Grasp Opt. & Rk. | 2 | 3.03 |
| **Total** | - | **6.51** |

Table 3: **Runtime breakdown.**

approaches (see Table 4). We anticipate that real-time feasibility can be pushed closer to real-time by adopting faster samplers [40, 41] and by increasing parallelism in the optimization stage.

| Method | Intermediate Rep. | Partial Obs. | Cluttered | Runtime (s) |
|---|---|---|---|---|
| GraspTTA | Contact Map | × | × | 43.23 |
| UniGrasp | 3 Contact Points | × | × | 9.33 |
| GenDexGrasp | Contact Map | × | × | 16.42 |
| **CADGrasp (Ours)** | **Sparse IBS** | ✓ | ✓ | **6.51** |

Table 4: **Runtime and capability comparison with representative two-stage methods.** Our approach is faster and enables dexterous grasping under partial observations in cluttered scenes, both of which are essential for real-world deployment.

### 4.5 Ablation Study.

We conduct comprehensive ablation studies on the dense scenes of GraspNet-1Billion test set to validate our design choices.

- **Module Interaction.** We first analyze the interaction effects between key modules in Table 5. The results show that removing any component leads to a performance drop, with the full model achieving the best success rate 86.5%. Notably, decomposing the contact representation ('Decompose') for the thumb and other fingers provides a substantial improvement (e.g., 86.5% vs. 56.1% without it). This confirms the critical role of the thumb in dexterous grasping [42] and validates our design to model it separately. Both the IBS and Grasp Pose Ranking modules are also proven effective.

| IBS Ranking | Grasp Pose Ranking | Decompose | Success Rate (%) |
|:---:|:---:|:---:|:---:|
| × | × | ✓ | 73.1 |
| × | ✓ | × | 53.8 |
| × | ✓ | ✓ | 83.9 |
| ✓ | × | × | 26.9 |
| ✓ | × | ✓ | 75.7 |
| ✓ | ✓ | × | 56.1 |
| ✓ | ✓ | ✓ | **86.5** |

Table 5: **Ablation study.** The interaction effects between key design elements in the dense-scene subset from the GraspNet-1Billon test set.

- **Voxel Resolution.** Next, we study the effect of voxel resolution in Table 6. We select a voxel size of 5mm to balance accuracy and efficiency. Finer resolutions (2.5mm) offer only marginal gains while significantly increasing memory usage, whereas coarser resolutions (10mm) lead to a clear performance drop.
- **Object Size.** Finally, we analyze the performance on objects of different sizes in Table 7. The results indicate that the success rate for small objects is indeed lower. However, combined with the findings in Table 6, we attribute this to the inherently higher precision required for grasping small objects, rather than insufficient voxel resolution.

| Voxel Size (mm) | Memory (GB) | SR (%) |
|:---:|:---:|:---:|
| 2.5 | 2.12 | 81.2 |
| **5** | **0.84** | **81.0** |
| 10 | 0.36 | 72.4 |

Table 6: **Ablation on voxel resolution.** Bolded items denote selected hyperparameters, balancing computational efficiency and performance.

| Volume Range ($m^3$) | SR (%) | Prop. (%) |
|:---:|:---:|:---:|
| (0, 0.00025) | 77.0 | 33.0 |
| [0.00025, 0.0005) | 78.8 | 36.9 |
| [0.0005, 0.001) | 82.6 | 22.3 |
| [0.001, 0.0015) | 82.3 | 6.0 |
| [0.0015, $+\infty$) | 91.5 | 1.9 |

Table 7: **Success rate vs. object volume.**

## 5 Conclusion

In this paper, we enhance general dexterous grasp pose prediction in cluttered scenes by proposing a two-stage framework. The first stage predicts our proposed compact, scene-decoupled, contact- and collision-aware intermediate representation, which serves as the target for the second-stage optimization. To ensure the quality of the predicted representation, we introduce an occupancy-diffusion model with voxel-level conditional guidance and force closure filtering. To generate stable and collision-free grasp poses, we further propose several energy functions and ranking strategies for pose optimization. Comprehensive simulation and real-world experiments demonstrate the effectiveness of our method.

**Limitations.** Although our method demonstrates better generalization, it primarily struggles with small objects, which could be addressed by incorporating more small objects during training. Additionally, the current second-stage optimization is time-consuming due to the use of the DDPM [43] for sampling IBS. This can be further optimized by adopting the DDIM [44].

## Acknowledgments and Disclosure of Funding

This work is supported by the National Natural Science Foundation of China - General Program (Project ID: 62376006), National Youth Talent Support Program (Project ID: 8200800081).

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

# Appendix

In this appendix, we first describe the implementation details in Section A, followed by additional experimental results in Section B.

## A  Implementation Details

### A.1  Training Details

We train our model on 8 NVIDIA RTX 4090 GPUs with a batch size of 64. We use the AdamW optimizer with a learning rate of $6e^{-5}$ and trained the model for 130 epochs. The training procedure takes about 2 days.

### A.2  Hyperparameters

The hyperparameters employed in our experiments are detailed in Table 8. The size of IBS volume is configured to $0.2\,\mathrm{m} \times 0.2\,\mathrm{m} \times 0.2\,\mathrm{m}$, adequately encompassing the interaction space between the dexterous hand and the object, while remaining sufficiently compact to focus on the critical local grasping region. The resolution of the IBS volume is set to $40 \times 40 \times 40$, striking a balance between computational efficiency and the accuracy of the IBS surface representation. Both the IBS sampling and grasp pose optimization processes are executed concurrently, with the number of IBS candidates and grasp poses each limited to 5. The weights in the contact energy $\mathbf{E}_d$ are meticulously adjusted to balance the contacts between the object and the thumb, as well as the other fingers, whereas the weights in the overall energy $\mathbf{E}$ are calibrated to harmonize the various energy terms. The hyperparameter for denoising timesteps is adopted from [35].

| Hyperparameter | Value |
|---|---|
| IBS Volume Size | $0.2m \times 0.2m \times 0.2m$ |
| IBS Resolution ($n$) | $40 \times 40 \times 40$ |
| Number of IBS Candidates ($m$) | 5 |
| Number of Grasp Poses ($k$) | 5 |
| Weights in Contact Energy $\mathbf{E}_d$ ($\alpha_1, \alpha_2, \alpha_3$) | 80, 100, 2 |
| Weights in Overall Energy $\mathbf{E}$ ($\lambda_1, \lambda_2, \lambda_3, \lambda_4$) | 5, 1, 1000, 1 |
| Denoising Timesteps | 50 |

Table 8: Hyperparameters used in our experiments.

## B  More Results

In this section, we provide additional results of our method to demonstrate the effectiveness, robustness, scalability, and generalization ability of our method.

### B.1  Qualitative Results

In this section, we present additional qualitative evaluations of our proposed method. Figure 7 showcases the perception results of our method in both simulated and real-world environments. The results clearly demonstrate that our approach successfully discerns the sparse IBS volume pertinent to a success grasp pose from a single-view point cloud in cluttered settings. It effectively differentiates the contact regions of the thumb and other fingers with the object. Furthermore, the second-stage optimization, guided by Sparse IBS constraints, adaptly refines collision-free and plausible grasp poses. Notably, the adoption of point cloud representation minimizes the sim-to-real gap, ensuring that our method can generalize to real-world settings without additional training, thereby achieving robust dexterous grasping capabilities.

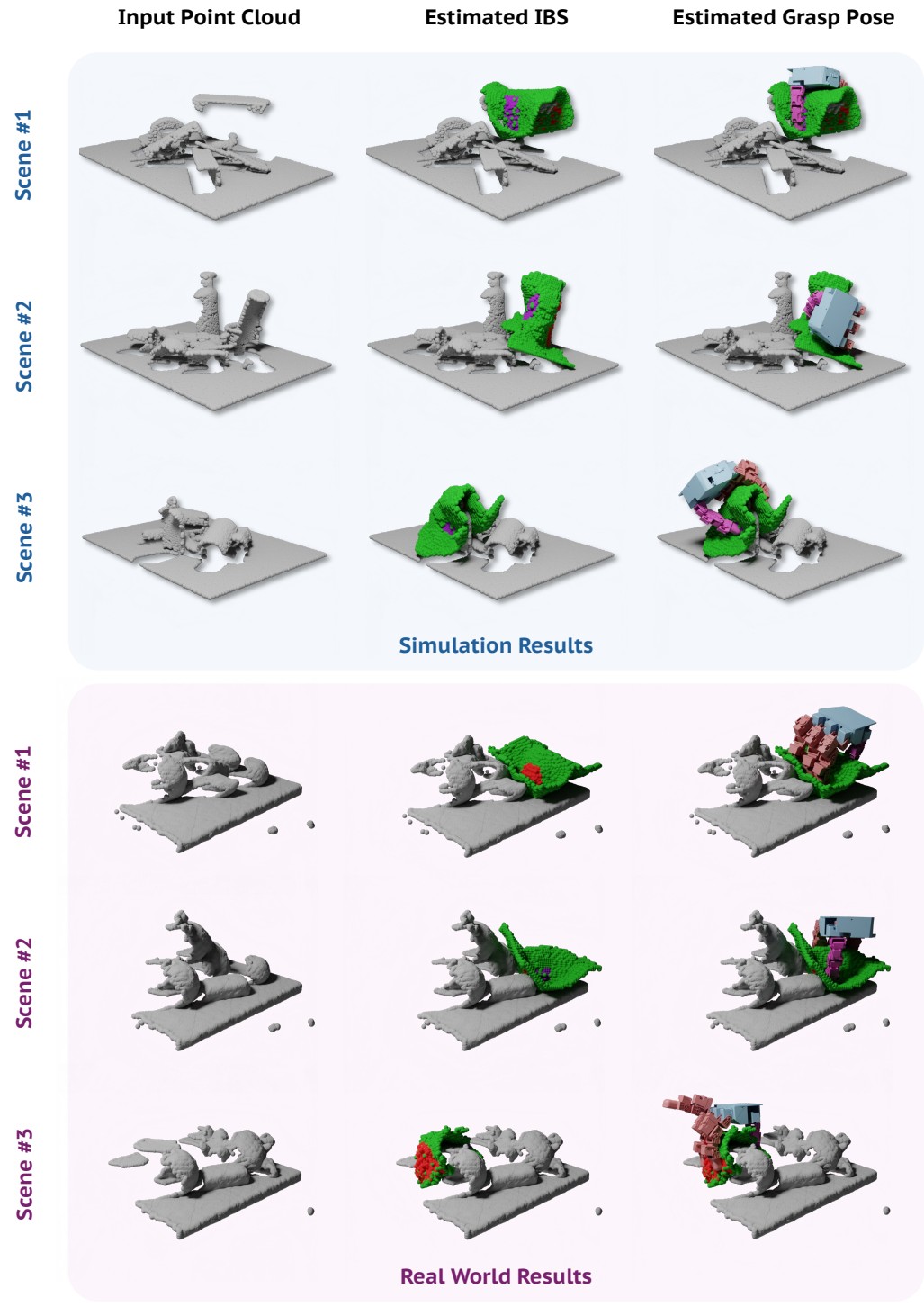

Figure 7: Qualitative results of our method in simulated (upper panel) and real-world environments (lower panel). From left to right: the initial single-view point cloud input, the sparse IBS prediction from the initial stage, and the optimized grasp pose from the subsequent stage. The purple areas indicate thumb contact, red areas denote contact by other fingers, and green areas represent non-contact regions on the IBS surface.

## B.2 Grasp Diversity Analysis

We analyze the distribution of joint configurations for all predicted grasp poses by **CADGrasp** and DexGraspNet2.0 [11] within the GraspNet-1B loose scenarios [11]. This analysis aimed to compare the diversity of grasp poses generated by the two methods. Taking the thumb as an example, Figure 8 illustrates that our method can generate diverse grasps with significantly higher dexterity compared to DexGraspNet2.0. This highlights our approach's capability to produce a wider range of effective grasp configurations, enhancing its applicability in complex scenarios.

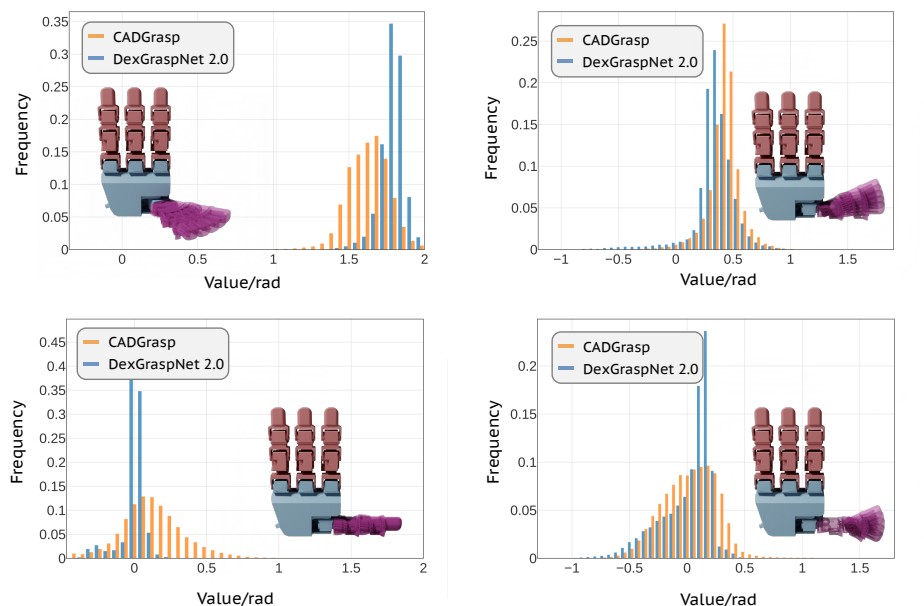

Figure 8: Grasp diversity analysis of **CADGrasp** and DexGraspNet2.0 within the GraspNet-1B loose scenarios. The histogram illustrates the distribution of joint configurations for the thumb.

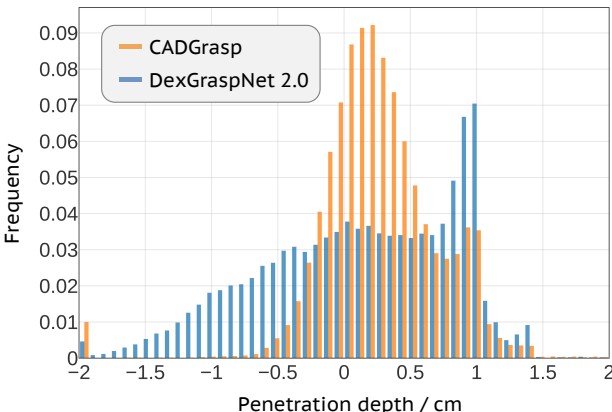

Figure 9: Penetration depth analysis of predicted grasp poses by **CADGrasp** and DexGraspNet2.0 within the GraspNet-1B loose scenarios. The histogram illustrates the distribution of maximal penetration depths.

## B.3 Grasp Quality Analysis

In evaluating grasp quality, the proximity and penetration between the predicted grasp pose and the object serve as crucial indicators. We analyzed the maximal penetration depth (in cm) of all predicted grasp poses by CADGrasp and DexGraspNet2.0 within the GraspNet-1B loose scenarios. This metric is defined as the maximal penetration depth from the object point cloud to the hand meshes. As illustrated in Figure 9, our method demonstrates a concentration of penetration depths near the object's surface (penetration depth = 0), attributed to the constraints imposed by the IBS Surface and contact points. This further underscores the superiority of our approach in achieving precise and effective grasping.

