# OpenReview forum: "CADGrasp: Learning Contact and Collision Aware General Dexterous Grasping in Cluttered Scenes"
_NeurIPS.cc/2025/Conference — NeurIPS 2025 poster_

### Official Review · Reviewer_EAK1 · 2025-06-22

**Clarity:** 3
**Significance:** 3
**Originality:** 3
**Rating:** 5
**Confidence:** 4

**Summary:**

The paper proposes CADGrasp, a two-stage algorithm to empower contact- and collision-aware dexterous grasping in cluttered scenes from a single-view point cloud. The first stage predicts an IBS surface in voxels and the second stage performs an energy-based optimization for grasp poses. The method has demonstrated strong performance against baselines in cluttered scenes both in simulation and real-world experiments and across two hand embodiments.

**Questions:**

1. Do you use simple motion planning to execute the pre-grasp?
2. What is the inference speed of the method?
3. "Additionally, the current second-stage optimization is time-consuming due to the use of the DDPM for sampling IBS." In my understanding, the second stage optimizes the hand pose given a fixed IBS surface. Why is the DDPM used for sampling IBS in the second-stage optimization?
4. How does the voxel resolution for IBS surface prediction affect the overall performance? Clearly, there is a trade-off between the precision and the computational cost.

**Ethical Concerns:**

["NO or VERY MINOR ethics concerns only"]

**Final Justification:**

The authors' response has addressed my concerns and questions. I would like to change my score to "Accept".

**Limitations:**

1. As mentioned by the authors, the optimization is time-consuming.
2. Voxel resolution can largely limit the precision of the IBS surface prediction. And computational cost grows cubically when we increase the resolution.
3. Small and thin objects are particularly subject to the voxelization.

**Paper Formatting Concerns:**

None.

**Quality:**

3

**Strengths And Weaknesses:**

Strengths:
1. The method effectively models the grasps in cluttered scenes with only a single-view point cloud that avoids collision with non-target objects and captures the contact information.
2. The method is object- and hand-agnostic so that it can generalize to different objects and hand embodiments.
3. The two stage design is modular and interpretable.
4. The high performance in real-world experiments provides solid evidence for the effectiveness of the method.

Weaknesses:
1. As mentioned by the authors, the optimization is time-consuming.
2. Voxel resolution can largely limit the precision of the IBS surface prediction. And computational cost grows cubically when we increase the resolution.

---

> ### Author Rebuttal · Authors · 2025-07-31
>
> We sincerely appreciate your constructive feedback and are excited that you find our work "effective", "interpretable", and "solid"! We want to clarify your concerns and questions in the following.
>
> > **Q1: As mentioned by the authors, the optimization is time-consuming. What is the inference speed of the method?**
>
> **A1:** We thank you for your comment. We provide the following clarifications and will include a discussion of this part in the final version.
>
> Thank you for raising this important point regarding the inference time of our method. In our experiments, the average inference runtime per sample on a single NVIDIA RTX 4090 GPU is summarized in Table 2. While our approach currently does not achieve real-time performance, we emphasize that these results are obtained with unoptimized code and no engineering-level acceleration. There is significant potential to reduce inference time using more efficient sampling strategies [1][2] and further parallelization. Importantly, even in its current implementation, our method achieves faster inference compared to prior state-of-the-art two-stage approaches (see Table 3), while also delivering strong grasping performance in challenging real-world scenarios. We will elaborate on the potential for further speedup and optimization in the final version.
>
> **Table 1: Average runtime per stage of our method (NVIDIA RTX 4090)**
> | Module                            | Stage                       | Average Runtime (seconds)   |
> |-----------------------------------|-----------------------------|-----------------------------|
> | Wrist Pose Estimation             | 1                           | 1.38                        |
> | Diffusion-based IBS Generation    | 1                           | 1.39                        |
> | IBS Ranking                       | 1                           | 0.71                        |
> | Grasp Pose Optimization & Ranking | 2                           | 3.03                        |
> | Total                             | -                           | 6.51                        |
>
> **Table 2: Comparison with existing two-stage methods**
> | Method              | Intermediate Representation | Partial Observation | Cluttered Scene | Runtime (seconds)           |
> |---------------------|-----------------------------|---------------------|-----------------|-----------------------------|
> | GraspTTA            | Contact Map                 | ✗                   | ✗               | 43.23                       |
> | UniGrasp            | 3 Contact Points            | ✗                   | ✗               | 9.33                        |
> | GenDexGrasp         | Contact Map                 | ✗                   | ✗               | 16.42                       |
> | **CADGrasp (Ours)** | **Sparse IBS**              | **✓**               | **✓**           | **6.51**                    |
>
> > **Q2: Voxel resolution can largely limit the precision of the IBS surface prediction. And computational cost grows cubically when we increase the resolution.**
>
> **A2:** We appreciate your comment and agree that voxel resolution is a key hyperparameter that impacts both the precision of IBS surface prediction and the associated computational cost. In our work, we select a voxel size of 5mm to balance accuracy and efficiency. This resolution is chosen because it is sufficient to capture contact surfaces for most objects in practical applications, while still maintaining manageable memory and runtime requirements (see Table 1 in A1 for runtime details). To further investigate this trade-off, we conducted ablation experiments on the GraspNet-1B Loose setting, as shown in Table 3 below. Finer resolutions lead to substantially higher memory consumption, while offering only marginal improvements in success rate. Conversely, coarser resolutions reduce memory usage but result in clear performance drops. We will add a more detailed discussion of this design choice in the final version of the paper.
>
> **Table 3: Ablation Study on Voxel Resolution**
> | Voxel Size (mm)  | Memory Usage (GB)  | SR (%) on Dense Scenes of GraspNet-1B |
> |------------------|--------------------|---------------------------------------|
> | 2.5              | 2.12               | 81.2                                  |
> | 5                | 0.84               | 81.0                                  |
> | 10               | 0.36               | 72.4                                  |
>
> > **Q3: Do you use simple motion planning to execute the pre-grasp?**
>
> **A3:** We thank the reviewer for this question. In our experiments, all methods employ the same motion planning algorithm to execute both pre-grasp and grasp poses. Developing goal-conditioned reinforcement learning algorithms for planning based on grasp poses is a promising direction for future work.
>
> > **Q4: The paper lacks a deeper theoretical analysis or visualization to explain why contact and collision conditioning improve grasp synthesis.**
>
> **A4:** We thank you for pointing out the importance of deeper theoretical analysis and visualization of our sparse IBS representation and the effect of contact and collision conditioning. We provide further clarification from two perspectives:
> - **Compared to one-stage direct grasp pose prediction:** Directly mapping 3D point cloud observations to grasp poses is challenging due to the highly non-linear nature of the mapping from 3D point cloud space to pose space, resulting in low sample efficiency[3][4]. Furthermore, such representations are usually embodiment-specific, limiting cross-embodiment generalization. Our sparse IBS representation is spatially aligned with the 3D point cloud, improving sample efficiency and enabling our two-stage approach to achieve cross-embodiment generalization, as shown in our experiments.
> - **Compared to two-stage methods that optimize from predicted contact map:** Approaches using contact map as intermediate representations generally assume that complete point cloud observations are available, which is unrealistic in real-world settings with partial observation. Our proposed sparse IBS representation, in contrast, does not rely on this assumption. As shown in both the quantitative experiments in the manuscript and qualitative visualizations in the supplementary material, our method achieves robust dexterous grasp pose prediction using only single-view partial point cloud inputs, making it suitable for complex, real-world tasks.
>
> We appreciate the suggestion and agree that further theoretical insights would strengthen our work. We will include additional theoretical analysis and visualizations in the final version.
>
> > **Q5: "Additionally, the current second-stage optimization is time-consuming due to the use of the DDPM for sampling IBS." In my understanding, the second stage optimizes the hand pose given a fixed IBS surface. Why is the DDPM used for sampling IBS in the second-stage optimization?**
>
> **A5:** We apologize for the confusion caused by this statement. This was a misstatement in the manuscript, and the intended statement should be: “The Diffusion-based IBS prediction in the first stage is time-consuming.” Thank you for catching this error. We will correct this in the final version.
>
> > **Q6: Small and thin objects are particularly subject to the voxelization.**
>
> **A6:** We appreciate your comment. As shown in Table 3 of A2, increasing the voxel resolution yields only marginal improvements in grasp success rate. However, we agree that object size has a substantial impact on grasping performance. In Table 4 below, we report the success rate of our method on the GraspNet-1B (Loose) setting as a function of object volume, along with the proportion of objects in each size category. The results indicate that, within a reasonable range, the grasp success rate for small objects is indeed lower compared to larger objects. Combined with the findings in Table 3 of A2, we attribute this to the inherently higher demands for precision and strategy when grasping small and thin objects, rather than to insufficient voxel resolution. We will add further discussion of this point in the final version.
>
> **Table 4: Success Rate on Small and Thin Objects**
> | Volume Range (m³) | Success Rate (%) | Proportion (%) |
> |-------------------|------------------|----------------|
> | (0, 0.00025)      | 77.0             | 33.0           |
> | [0.00025, 0.0005) | 78.8             | 36.9           |
> | [0.0005, 0.001)   | 82.6             | 22.3           |
> | [0.001, 0.0015)   | 82.3             | 6.0            |
> | [0.0015, +∞)      | 91.5             | 1.9            |
>
>
> [1] Salimans T, et al. Progressive distillation for fast sampling of diffusion models. ICLR 2022.
>
> [2] Song Y, et al. Consistency models. ICML 2023.
>
> [3] Haoming Li, et al. Contact2Grasp: 3D Grasp Synthesis via Hand-Object Contact Constraint. IJCAI 2023.
>
> [4] Hao-Shu Fang, et al. AnyDexGrasp: Learning General Dexterous Grasping for Different Hands with Human-level Efficiency. ArXiv 2025.

---

> > ### Comment · Reviewer_EAK1 · 2025-08-02
> >
> > Thank you for your clarifications and the additional benchmarking results.
> >
> > 1. The ablation study on voxel size is informative. Voxel resolution of 5mm seems to strike a good balance between precision and computational cost regular sized objects.
> > 2. The benchmarking results alleviate my concerns regarding the method's speed for robotic tasks. Compared to other optimization-based approaches, the runtime is quite reasonable. In the field, optimization-based methods combined with motion planning are often somewhat isolated from end-to-end learning approaches. I look forward to future work that explores ways to bridge this gap and leverage the strengths of both paradigms.
> >
> > I will keep my original score at this stage.

---

> > > ### Author Response · Authors · 2025-08-04
> > >
> > > Thank you for your thoughtful feedback. We are pleased that our clarifications and additional results address your concerns. We also agree that combining grasp pose generation with end-to-end learning approaches holds significant potential for future research. We sincerely appreciate your constructive comments and insights.

---

### Official Review · Reviewer_s4Sp · 2025-07-01

**Clarity:** 3
**Significance:** 3
**Originality:** 3
**Rating:** 4
**Confidence:** 5

**Summary:**

This paper proposes CADGrasp, a diffusion-based framework for dexterous grasp synthesis that explicitly integrates contact awareness and collision avoidance. The method is trained on a large-scale synthetic dataset and evaluated on both simulated and real-world grasping tasks. CADGrasp outperforms baseline methods in simulation.

**Questions:**

1. Could you provide more comprehensive ablations, including “with IBS ranking-only”, “with decompose-only”, and “with grasp pose ranking-only” settings to clarify the joint impact of each component?
2. Can the number of trials in real-world experiments be standardized across all methods to ensure fair comparison?
3. How robust is CADGrasp to noisy supervision during training, especially in contact or collision labels?
4.Is CADGrasp feasible without object meshes during inference? If not, how practical is this in real-world grasping scenarios?

**Ethical Concerns:**

["NO or VERY MINOR ethics concerns only"]

**Final Justification:**

The authors' rebuttals adequately addressed my concerns. I suggest borderline accept of the paper.

**Limitations:**

1. The ablation study is limited. It only removes individual components (e.g., IBS ranking or decompose), without evaluating combinations, making it difficult to assess interaction effects between modules.
2. Real-world evaluations in Table 3 show only marginal improvement over baselines, with inconsistent numbers of trials across methods, which affects the fairness and reliability of the comparison.
3. The paper lacks deeper theoretical analysis or visualization to explain why contact and collision conditioning improves grasp synthesis.

**Quality:**

2

**Strengths And Weaknesses:**

Strengths
1. The paper proposes a novel diffusion-based grasp generation framework that incorporates both contact and collision awareness, addressing a critical limitation in existing dexterous grasping methods.
2. The formulation is technically sound, and the system demonstrates generalization to unseen and real-world scenes without additional fine-tuning.
3. The authors provide simulation and real-robot experiments, showing applicability to practical settings.

Weaknesses
1. The ablation study is limited. It only removes individual components (e.g., IBS ranking or decompose), without evaluating combinations, making it difficult to assess interaction effects between modules.
2. Real-world evaluations in Table 3 show only marginal improvement over baselines, with inconsistent numbers of trials across methods, which affects the fairness and reliability of the comparison.
3. The paper lacks deeper theoretical analysis or visualization to explain why contact and collision conditioning improves grasp synthesis.

---

> ### Author Rebuttal · Authors · 2025-07-31
>
> We sincerely appreciate your constructive feedback and are excited that you find our work "novel", "addressing a critical limitation in existing dexterous grasping methods", and "technically sound"! We want to clarify your concerns and questions in the following.
>
> > **Q1: The ablation study is limited. It only removes individual components (e.g., IBS ranking or decompose), without evaluating combinations, making it difficult to assess interaction effects between modules.**
>
> **A1:** We thank you for your suggestions regarding more comprehensive ablation studies. In response, we conducted additional experiments to analyze the interaction effects between different modules and the contribution of each design choice (see Table 1 below). The results show that decomposed modeling of thumb and other finger contacts with the object leads to a substantial improvement in grasp performance, confirming that the thumb plays a crucial role in dexterous grasping [1]. Additionally, both the IBS ranking and grasp pose ranking modules further enhance overall grasp performance. We will update Table 3 in the final version and provide further discussion in the manuscript.
>
> **Table 1: Ablation Study on Interaction Effects and Design Choices**
> | IBS Ranking | Grasp Pose Ranking | Decompose | SR (%) on Dense Scenes of GraspNet-1B |
> |-------------|--------------------|-----------|------------------|
> | ✗           | ✗                  | ✓         | 73.1             |
> | ✗           | ✓                  | ✗         | 53.8             |
> | ✗           | ✓                  | ✓         | 83.9             |
> | ✓           | ✗                  | ✗         | 26.9             |
> | ✓           | ✗                  | ✓         | 75.7             |
> | ✓           | ✓                  | ✗         | 56.1             |
> | ✓           | ✓                  | ✓         | 86.5             |
>
>
> > **Q2: Real-world evaluations in Table 3 show only marginal improvement over baselines, with inconsistent numbers of trials across methods, which affects the fairness and reliability of the comparison.**
>
> **A2:** We apologize for any confusion. As described in the Real-world Setup section of the manuscript, we follow the cleared-table grasping evaluation protocol used in GS-Net [2]. For each scene, the policy continues grasping until two consecutive failures occur. As some baselines tend to reach this failure criterion earlier, the total number of grasp attempts appears inconsistent across methods in Table 3. This actually reflects the greater robustness of our method. We will clarify this evaluation protocol and the reason for varying trial numbers in more detail in the final version for better clarity.
>
> > **Q3: The paper lacks a deeper theoretical analysis or visualization to explain why contact and collision conditioning improve grasp synthesis.**
>
> **A3:** We thank you for pointing out the importance of deeper theoretical analysis and visualization of our sparse IBS representation and the effect of contact and collision conditioning. We provide further clarification from two perspectives:
> - **Compared to one-stage direct grasp pose prediction:** Directly mapping 3D point cloud observations to grasp poses is challenging due to the highly non-linear nature of the mapping from 3D point cloud space to pose space, resulting in low sample efficiency[3][4]. Furthermore, such representations are usually embodiment-specific, limiting cross-embodiment generalization. Our sparse IBS representation is spatially aligned with the 3D point cloud, improving sample efficiency and enabling our two-stage approach to achieve cross-embodiment generalization, as shown in our experiments.
> - **Compared to two-stage methods that optimize from predicted contact map:** Approaches using the contact map as intermediate representations generally assume that complete point cloud observations are available, which is unrealistic in real-world settings with partial observation. Our proposed sparse IBS representation, in contrast, does not rely on this assumption. As shown in both the quantitative experiments in the manuscript and qualitative visualizations in the supplementary material, our method achieves robust dexterous grasp pose prediction using only single-view partial point cloud inputs, making it suitable for complex, real-world tasks.
>
> We appreciate the suggestion and agree that further theoretical insights would strengthen our work. We will include additional theoretical analysis and visualizations in the final version.
>
> > **Q4: How robust is CADGrasp to noisy supervision during training, especially in contact or collision labels?**
>
> **A4:** We appreciate the reviewer’s question. In our training process, the contact and collision labels are not manually annotated but are generated automatically in simulation. Specifically, force-closure grasps are optimized in simulation, and both the IBS surface and contact ground-truth are computed programmatically. This approach ensures label consistency and reduces the likelihood of significant supervision noise.
>
> > **Q5: Is CADGrasp feasible without object meshes during inference? If not, how practical is this in real-world grasping scenarios?**
>
> **A5:** Unlike contact map-based two-stage methods, our model does not require object CAD models or complete point cloud observations as input, thanks to our proposed sparse IBS representation. CADGrasp only relies on single-view partial point clouds, enabling dexterous grasping in complex real-world scenarios using partial observations. This design ensures that our method can be directly deployed in real-world environments without any special modifications or assumptions.
>
> [1] Thomas Feix, et al. The grasp taxonomy of human grasp types. IEEE Transactions on human-machine systems, 2015.
>
> [2] Chenxi Wang, et al. Graspness discovery in clutters for fast and accurate grasp detection. ICCV 2021.
>
> [3] Haoming Li, et al. Contact2Grasp: 3D Grasp Synthesis via Hand-Object Contact Constraint. IJCAI 2023.
>
> [4] Hao-Shu Fang, et al. AnyDexGrasp: Learning General Dexterous Grasping for Different Hands with Human-level Efficiency. ArXiv 2025.

---

> > ### Comment · Reviewer_s4Sp · 2025-08-04
> > **Official Comment by Reviewer s4Sp**
> >
> > The rebuttal clarifies my questions, particularly with additional experiments. I hope the authors can incorporate the rebuttal materials into the paper to enhance the paper quality.

---

> > > ### Author Response · Authors · 2025-08-05
> > >
> > > Dear Reviewer **#s4Sp**,
> > >
> > >
> > > Thank you for your feedback and for acknowledging the clarifications and additional experiments in our rebuttal. We appreciate your suggestions and will incorporate these materials to enhance the quality of the paper.
> > >
> > > If you have any further questions, please let us know, and we will be happy to address them.
> > >
> > >
> > > Best regards,
> > >
> > > The Authors

---

> > > > ### Comment · Reviewer_s4Sp · 2025-08-07
> > > > **Official Comment by Reviewer s4Sp**
> > > >
> > > > Thank you for the thorough rebuttal, and I am maintaining my positive rating.

---

### Official Review · Reviewer_H1AQ · 2025-07-03

**Clarity:** 3
**Significance:** 3
**Originality:** 3
**Rating:** 4
**Confidence:** 3

**Summary:**

CADGrasp tackles dexterous grasping in highly cluttered scenes by splitting the problem into two stages. Stage 1 is representation prediction. A conditional occupancy-diffusion model, guided voxel-wise by the canonicalised scene point cloud, predicts a sparse Interaction-Bisector Surface (IBS) that is both contact-aware and collision-aware. Stage 2 is for pose optimization. Several energy terms are minimised to recover a full 25-DoF hand pose that realises the predicted sparse IBS. Extensive experiments on 670 simulated cluttered scenes (≈1300 objects) and on five real scenes with 30 diverse household objects show higher success rates than regression and diffusion baselines.

**Questions:**

1. Could the authors report standard deviations over multiple random scene orderings?
2. Could the authors provide average test time for Stage 1 sampling (per IBS candidate) and Stage 2 optimisation (per initialisation) on the hardware used. if possible, the authors are also encouraged to discuss feasibility for real-time grasp planning.
3. Will the implementation and trained models be open-sourced upon acceptance, and if so under which licence?

**Ethical Concerns:**

["NO or VERY MINOR ethics concerns only"]

**Final Justification:**

I will maintain the original score.

**Limitations:**

The paper discusses the limitations of the work performed by the authors yet not directly addresses it.

**Paper Formatting Concerns:**

No major formatting issues in this paper.

**Quality:**

3

**Strengths And Weaknesses:**

**Strengths**
1. IBS is a conceptually well-motivated intermediate representation. By predicting a sparse IBS that is decoupled from complete scene geometry, the method avoids the brittle requirement for full object reconstruction.
2. CADGrasp outperforms state-of-the-art baselines across loose, random and dense clutter, and maintains competitive accuracy on unseen ShapeNet objects.
3. The provided supplementary video organizes the contents clearly.
4. The paper explicitly explores the benefit of each design choice.

**Weakness**
1. All success-rate tables should provide variance across random seed.
2. The diffusion sampling plus gradient-based optimisation could be slow; yet runtimes per grasp are absent.
3. Comparison to some recent contact-based two-stage graspers in clutter may be ommited, such as UniDexGrasp++ and UniDexGrasp in cluttered settings.
4. Some typos are found. For example, in line 296, the "." should be ",".

---

> ### Author Rebuttal · Authors · 2025-07-31
>
> We sincerely appreciate your constructive feedback and are excited that you find our work "well-motivated", "competitive", and "clear"! We want to clarify your concerns and questions in the following.
> > **Q1: All success-rate tables should provide variance across random seed.**
>
> **A1:** We thank you for your comment, and we agree that variance is important for evaluating the robustness of our method. As suggested, we conducted each experiment with 20 different random seeds and report the success rates together with the corresponding standard deviations in Table 1 below. The observed standard deviations are consistently small across different datasets and scene types, which indicates that our method exhibits stable performance across random initializations. Note that the reported variance captures both the inherent randomness of our method and the variability introduced by the simulation environment, further demonstrating the robustness of our approach.
>
> **Table 1: Success rates of CADGrasp on different datasets and scene types**
> | Dataset     | Scene Type   | Success Rate (%) ± Std. Dev. |
> |-------------|--------------|------------------------------|
> | GraspNet-1B | Dense        | 86.0 ± 2.2                   |
> | GraspNet-1B | Random       | 85.0 ± 3.1                   |
> | GraspNet-1B | Loose        | 80.1 ± 2.1                   |
> | ShapeNet    | Dense        | 78.6 ± 2.5                   |
> | ShapeNet    | Random       | 77.0 ± 3.8                   |
> | ShapeNet    | Loose        | 75.0 ± 4.0                   |
>
>
>
> > **Q2: The diffusion sampling plus gradient-based optimisation could be slow; yet runtimes per grasp are absent.**
>
> **A2:** We thank you for your comment. We provide the following clarifications and will include a discussion of this part in the final version.
>
> We performed 50 inference runs on a single NVIDIA RTX 4090 GPU and report the average runtime for each stage of our method in Table 2 below. Although our method does not meet real-time requirements, we believe this is reasonable given its high performance and the potential of the proposed novel representation for cross-embodiment generalization. We also note that these reported runtimes are obtained without any engineering-level optimization—real-time feasibility can be further improved by using faster samplers [1][2] and by increasing parallelization in the second stage. Furthermore, as shown in Table 3, even without engineering acceleration, our method outperforms the speed of existing SOTA two-stage methods and provides effective representations for complex real-world grasping tasks.
>
> **Table 2: Average runtime per stage of our method (NVIDIA RTX 4090)**
> | Module                                               | Stage                       | Average Runtime (seconds)   |
> |------------------------------------------------------|-----------------------------|-----------------------------|
> | Wrist Pose Estimation                                | 1                           | 1.38                        |
> | Diffusion-based IBS Generation (5 Candidates)        | 1                           | 1.39                        |
> | IBS Ranking                                          | 1                           | 0.71                        |
> | Grasp Pose Optimization (5 Initialisation) & Ranking | 2                           | 3.03                        |
> | Total                                                | -                           | 6.51                        |
>
> **Table 3: Comparison with existing two-stage methods**
> | Method              | Intermediate Representation | Partial Observation | Cluttered Scene | Runtime (seconds)           |
> |---------------------|-----------------------------|---------------------|-----------------|-----------------------------|
> | GraspTTA            | Contact Map                 | ✗                   | ✗               | 43.23                       |
> | UniGrasp            | 3 Contact Points            | ✗                   | ✗               | 9.33                        |
> | GenDexGrasp         | Contact Map                 | ✗                   | ✗               | 16.42                       |
> | **CADGrasp (Ours)** | **Sparse IBS**              | **✓**               | **✓**           | **6.51**                    |
>
>
> > **Q3: Comparison to some recent contact-based two-stage graspers in clutter may be omitted, such as UniDexGrasp++ and UniDexGrasp in cluttered settings.**
>
> **A3:** We thank you for your comment. We would like to clarify the following points:
> - We would like to clarify that UniDexGrasp and UniDexGrasp++ focus on single-object grasping and do not support grasping in cluttered scenes. Furthermore, these approaches have not been extended to the dexterous grasping in the real-world setting that is the focus of our work.
>
> - Existing contact-based two-stage methods, which first predict a contact map and then optimize for a grasp pose, generally rely on two strong assumptions: (1) access to full point clouds, and (2) single-object grasping. Both assumptions are unrealistic in the context of general real-world grasping in cluttered scenes. Nevertheless, following the modifications suggested by DexGraspNet 2.0, we adapted these two-stage baselines (e.g., GraspTTA in Table 1 of the manuscript) for a fair comparison under our proposed setting. Our method still outperforms these baselines in both performance and speed. Please refer to Table 2 in A2 for detailed runtime results. This advantage largely stems from our proposed sparse IBS representation, which does not require a complete scene point cloud.
>
> > **Q4: Some typos are found. For example, in line 296, the "." should be ",".**
>
> **A4:** We appreciate your careful reading of our manuscript. We will correct the typos in the final version.
>
> > **Q5: Will the implementation and trained models be open-sourced upon acceptance, and if so, under which licence?**
>
> **A5:** Yes, we plan to open-source the implementation and trained models upon acceptance to facilitate reproducibility and further research in this area. They will be released under the MIT License.
>
> [1] Salimans T, et al. Progressive distillation for fast sampling of diffusion models. ICLR 2022.
>
> [2] Song Y, et al. Consistency models. ICML 2023.

---

> ### Author Response · Authors · 2025-08-05
>
> Dear Reviewer **#H1AQ**,
>
> Thank you for your careful review. We sincerely appreciate your feedback and the time you dedicated to evaluating our work.
>
> Should you have any further questions or require additional clarification, please do not hesitate to contact us.
>
> Best regards,
>
> The Authors

---

### Official Review · Reviewer_vrT5 · 2025-07-06

**Clarity:** 3
**Significance:** 3
**Originality:** 3
**Rating:** 5
**Confidence:** 3

**Summary:**

This paper presents CADGrasp, a two-stage framework for dexterous grasping in cluttered scenes using single-view point cloud inputs. The approach introduces sparse IBS (Interaction Bisector Surface) as a contact and collision aware intermediate representation that serves as an optimization target. The first stage uses an occupancy-diffusion model to predict sparse IBS from point clouds, while the second stage optimizes grasp poses using energy functions based on the predicted IBS.

**Questions:**

**Representation Design Justification**: Could you provide more theoretical or empirical analysis of why the specific three-channel design for sparse IBS is optimal? Have you experimented with alternative representations?

**Computational Efficiency and Scalability**: Can you provide detailed timing analysis comparing your method to baselines? Given the two-stage nature with diffusion sampling and optimization, how does this scale to real-time applications?

**Generalization Boundaries**: While cross-embodiment results are impressive, what are the limits of this generalization? How sensitive is the method to significant differences in hand morphology, scale, or degrees of freedom? Could you provide analysis of failure cases where the representation breaks down?

**Ethical Concerns:**

["NO or VERY MINOR ethics concerns only"]

**Limitations:**

Yes

**Quality:**

3

**Strengths And Weaknesses:**

**Strength**

**Novel Intermediate Representation**: The sparse IBS representation is a compelling contribution that addresses the core challenge of cluttered scene grasping. By encoding geometric relationships and contact information in a scene-decoupled manner, it enables optimization without requiring complete object geometry—a significant advantage over existing two-stage methods.

**Strong Experimental Validation**: The paper provides comprehensive evaluation across 670 diverse cluttered scenes, demonstrating consistent improvements over strong baselines. The real-world experiments with 93.3% success rate provide convincing evidence of practical applicability.

**Cross-Embodiment Generalization**: The method's ability to generalize across different hand embodiments without retraining is noteworthy and demonstrates the universality of the proposed representation.

**Technical Soundness**: The two-stage framework is well-motivated and technically sound. The occupancy-diffusion model with voxel-level conditional guidance is appropriately designed for the high-dimensional IBS prediction task.

**Weaknesses**

**Limited Theoretical Analysis**: The paper lacks theoretical justification for why sparse IBS should be more effective than alternative representations. While empirical results are strong, deeper analysis of the representation's properties would strengthen the contribution.

**Computational Efficiency Concerns**: The method requires DDPM sampling for IBS generation followed by gradient-based optimization, which the authors acknowledge is time-consuming. The paper doesn't provide detailed timing comparisons or discuss scalability to real-time applications.

**Incomplete Ablation Studies**: While the paper includes ablation studies, some design choices lack sufficient justification. For example, the choice of three channels for sparse IBS (surface occupancy, thumb contact, other finger contact) could benefit from more systematic analysis.

---

> ### Author Rebuttal · Authors · 2025-07-31
>
> We sincerely appreciate your constructive feedback and are excited that you find our work "novel", "strong experimental validation", and "technical soundness"! We want to clarify your concerns and questions in the following.
>
> > **Q1: Limited Theoretical Analysis: The paper lacks a theoretical justification for why sparse IBS should be more effective than alternative representations. While empirical results are strong, a deeper analysis of the representation's properties would strengthen the contribution.**
>
> **A1:** We thank you for pointing out the importance of deeper theoretical analysis. We provide further clarification from two perspectives:
> - **Compared to one-stage direct grasp pose prediction:** Directly mapping 3D point cloud observations to grasp poses is challenging due to the highly non-linear nature of the mapping from 3D point cloud space to pose space, resulting in low sample efficiency[1][2]. Furthermore, such representations are usually embodiment-specific, limiting cross-embodiment generalization. Our sparse IBS representation is spatially aligned with the 3D point cloud, improving sample efficiency and enabling our two-stage approach to achieve cross-embodiment generalization, as shown in our experiments.
> - **Compared to two-stage methods that optimize from predicted contact map:** Approaches using the contact map as intermediate representations generally assume that complete point cloud observations are available, which is unrealistic in real-world settings with partial observation. Our proposed sparse IBS representation, in contrast, does not rely on this assumption. As shown in both the quantitative experiments in the manuscript and qualitative visualizations in the supplementary material, our method achieves robust dexterous grasp pose prediction using only single-view partial point cloud inputs, making it suitable for complex, real-world tasks.
>
> We appreciate your suggestions and agree that further theoretical insights would strengthen our work. We will include additional theoretical analysis and visualizations in the final version.
>
> > **Q2: Computational Efficiency Concerns: The method requires DDPM sampling for IBS generation followed by gradient-based optimization, which the authors acknowledge is time-consuming. The paper doesn't provide detailed timing comparisons or discuss scalability to real-time applications.**
>
> **A2:** We appreciate your comment. We provide the following clarifications and will include a discussion of this in the final version.
> - **Runtimes:**
> We thank the reviewers for highlighting the importance of computational efficiency. We performed 50 inference runs on a single NVIDIA RTX 4090 GPU and report the average runtime for each stage of our method in Table 1 below. Though our method does not meet real-time requirements, we believe this is reasonable given its high performance and the potential of the proposed novel representation for cross-embodiment generalization. We would also like to note that the reported runtime is without engineering-level optimization—real-time feasibility can be further improved by using faster samplers[3][4] and increasing parallelization in the second stage.
>
>     **Table 1: Average runtime per stage of our method (NVIDIA RTX 4090)**
>     | Module                            | Stage                       | Average Runtime (seconds)   |
>     |-----------------------------------|-----------------------------|-----------------------------|
>     | Wrist Pose Estimation             | 1                           | 1.38                        |
>     | Diffusion-based IBS Generation    | 1                           | 1.39                        |
>     | IBS Ranking                       | 1                           | 0.71                        |
>     | Grasp Pose Optimization & Ranking | 2                           | 3.03                        |
>     | Total                             | -                           | 6.51                        |
>
> - **Comparison with Baselines:**
> We also compare overall runtime with mainstream two-stage methods in Table 2. Our approach not only provides effective representations for complex real-world grasping but also achieves better runtime than baselines, even without inference-time optimizations.
>
>     **Table 2: Comparison with existing two-stage methods**
>     | Method              | Intermediate Representation | Partial Observation | Cluttered Scene | Runtime (seconds)           |
>     |---------------------|-----------------------------|---------------------|-----------------|-----------------------------|
>     | GraspTTA            | Contact Map                 | ✗                   | ✗               | 43.23                       |
>     | UniGrasp            | 3 Contact Points            | ✗                   | ✗               | 9.33                        |
>     | GenDexGrasp         | Contact Map                 | ✗                   | ✗               | 16.42                       |
>     | **CADGrasp (Ours)** | **Sparse IBS**              | **✓**               | **✓**           | **6.51**                    |
>
>
> > **Q3: Incomplete Ablation Studies: While the paper includes ablation studies, some design choices lack sufficient justification. For example, the choice of three channels for sparse IBS (surface occupancy, thumb contact, other finger contact) could benefit from more systematic analysis.**
>
> **A3:** We thank you for your suggestions regarding more comprehensive ablation studies. In response, we conducted additional experiments to analyze the interaction effects between different modules and the contribution of each design choice (see Table 3 below). The results show that decomposed modeling of thumb and other finger contacts with the object leads to a substantial improvement in grasp performance, confirming that the thumb plays a crucial role in dexterous grasping [5]. Additionally, both the IBS ranking and grasp pose ranking modules further enhance overall grasp performance. We will update Table 3 in the final version and provide further discussion in the manuscript.
>
> **Table 3: Ablation Study on Interaction Effects and Design Choices**
> | IBS Ranking | Grasp Pose Ranking | Decompose | SR (%) on Dense Scenes of GraspNet-1B |
> |-------------|--------------------|-----------|------------------|
> | ✗           | ✗                  | ✓         | 73.1             |
> | ✗           | ✓                  | ✗         | 53.8             |
> | ✗           | ✓                  | ✓         | 83.9             |
> | ✓           | ✗                  | ✗         | 26.9             |
> | ✓           | ✗                  | ✓         | 75.7             |
> | ✓           | ✓                  | ✗         | 56.1             |
> | ✓           | ✓                  | ✓         | 86.5             |
>
> > **Q4: Generalization Boundaries: While cross-embodiment results are impressive, what are the limits of this generalization? How sensitive is the method to significant differences in hand morphology, scale, or degrees of freedom? Could you provide an analysis of failure cases where the representation breaks down?**
>
> **A4:** We thank you for this insightful question. We agree that understanding the boundaries of cross-embodiment generalization is an important direction for future work. While our experiments demonstrate that the proposed sparse IBS representation enables a degree of cross-embodiment generalization, we do not over-claim this capability in the paper. As currently designed, the sparse IBS generation stage does not condition on the specific hand morphology, which means that generalization is primarily observed between dexterous hands with similar shape, scale, and degrees of freedom. For example, if the target hand is much smaller, when the fingertips satisfy the predicted contact constraints, the palm will collide with the object due to the shorter fingers. Similarly, hands with fewer degrees of freedom or very different structures cannot achieve the predicted contacts and will face kinematic infeasibility. Incorporating hand morphology and kinematics as part of the conditioning input for sparse IBS generation could potentially address these limitations and enable broader generalization—a direction we plan to explore in future work. We will discuss these boundaries and potential improvements more explicitly in the final version.
>
>
> [1] Haoming Li, et al. Contact2Grasp: 3D Grasp Synthesis via Hand-Object Contact Constraint. IJCAI 2023.
>
> [2] Hao-Shu Fang, et al. AnyDexGrasp: Learning General Dexterous Grasping for Different Hands with Human-level Efficiency. ArXiv 2025.
>
> [3] Salimans T, et al. Progressive distillation for fast sampling of diffusion models. ICLR 2022.
>
> [4] Song Y, et al. Consistency models. ICML 2023.
>
> [5] Thomas Feix, et al. The grasp taxonomy of human grasp types. IEEE Transactions on human-machine systems, 2015.

---

> > ### Comment · Reviewer_vrT5 · 2025-08-05
> >
> > Thanks for your comprehensive and thoughtful rebuttal, which addressed my concerns clearly and thoroughly.
> >
> > The authors provided additional theoretical justification for the use of sparse IBS, detailed runtime comparisons demonstrating computational efficiency, extended ablation studies supporting your network design, and a discussion of the boundaries of cross-embodiment generalization.
> >
> > These clarifications have reinforced my positive assessment of CADGrasp, and I will maintain my current score.

---

> > > ### Author Response · Authors · 2025-08-05
> > >
> > > Dear Reviewer **#vrT5**
> > >
> > > Thank you very much for your constructive feedback. We are glad that our additional explanations and experiments have addressed your concerns.
> > >
> > > If you have any further questions or need additional clarification, please feel free to let us know.
> > >
> > > Best regards,
> > >
> > > The Authors

---

### Decision · Program_Chairs · 2025-09-17

**Decision:**

Accept (poster)

**Comment:**

This paper proposes CADGrasp, a two-stage framework for dexterous grasping in cluttered scenes that introduces sparse Interaction Bisector Surface (IBS) as a novel intermediate representation to decouple contact and collision awareness from complete scene geometry. Its key strengths include: (1) the innovative sparse IBS representation that enables robust optimization without complete object geometry, (2) strong cross-embodiment generalization without retraining, and (3) exceptional real-world performance (93.3% success rate) across diverse clutter configurations.

The authors did a great rebuttal, and all reviewers are satisfied. The AC concurs with reviewers that the representation learning contribution is significant and the real-world validation compelling, and thus the final decision is accept. The authors are encouraged to include the additional feedback provided during the rebuttal and discussion period.